# Under-Actuated Motion Control of Haidou-1 ARV Using Data-Driven, Model-Free Adaptive Sliding Mode Control Method

**DOI:** 10.3390/s24113592

**Published:** 2024-06-02

**Authors:** Jixu Li, Yuangui Tang, Hongyin Zhao, Jian Wang, Yang Lu, Rui Dou

**Affiliations:** 1State Key Laboratory of Robotics, Shenyang Institute of Automation, Chinese Academy of Sciences, Shenyang 110016, China; lijixu@sia.cn (J.L.); zhaohongyin@sia.cn (H.Z.); wangjian3@sia.cn (J.W.); luyang1@sia.cn (Y.L.); dourui@sia.cn (R.D.); 2Institutes for Robotics and Intelligent Manufacturing, Chinese Academy of Sciences, Shenyang 110169, China; 3Key Laboratory of Marine Robotics, Liaoning Province, Shenyang 110169, China; 4University of Chinese Academy of Sciences, Beijing 100049, China

**Keywords:** Haidou-1 ARV, motion control, MFASMC method, under-actuated motion control, uncertain influence

## Abstract

We propose a data-driven, model-free adaptive sliding mode control (MFASMC) approach to address the Haidou-1 ARV under-actuated motion control problem with uncertainties, including external disturbances and parameter perturbations. Firstly, we analyzed the two main difficulties in the motion control of Haidou-1 ARV. Secondly, in order to address these problems, a MFASMC control method was introduced. It is combined by a model-free adaptive control (MFAC) method and a sliding mode control (SMC) method. The main advantage of the MFAC method is that it relies only on the real-time measurement data of an ARV instead of any mathematical modeling information, and the SMC method guarantees the MFAC method’s fast convergence and low overshooting. The proposed MFASMC control method can maneuver Haidou-1 ARV cruising at the desired forward speed, heading, and depth, even when the dynamic parameters of the ARV vary widely and external disturbances exist. It also addresses the problem of under-actuated motion control for the Haidou-1 ARV. Finally, the simulation results, including comparisons with a PID method and the MFAC method, demonstrate the effectiveness of our proposed method.

## 1. Introduction

Underwater robotic vehicles (URVs) are widely used in the ocean engineering field. Usually, we can classify them into remotely operated underwater vehicles (ROVs) [1] and autonomous underwater vehicles (AUVs) [2]. Over the last few decades, AUVs have been widely used in various marine activities, such as data collection [3,4], mapping of underwater topography [5], long oil pipe inspection [6], and even naval mine detection [7]. An autonomous and remotely-operated vehicle (ARV) is a new type of hybrid underwater vehicle that combines the characteristics of an autonomous underwater vehicle (AUV) and remotely controlled vehicle (ROV) [8]. In contrast to a regular AUV or ROV, an ARV can not only conduct large-scale underwater surveys but also conduct precise investigations and operations in specific areas without any mechanical structure change. A typical ARV survey usually follows an “autonomously search—land on and sampling—autonomously search” paradigm [9].

Haidou-1 ARV [10] is a full-ocean unmanned underwater vehicle capable of diving to 11,000 m. It was designed by Shenyang Institute of Automation, CAS, and is shown in Figure 1. It has dived up to 10,908 m and completed scientific expeditions to the Challenger Deep in the Mariana Trench in 2020 and 2021.

In order to accomplish particular tasks successfully, motion control technology is a very critical technology for Haidou-1 ARV. However, the motion control of the Haidou-1 ARV is challenging for the following reasons: (1) It is very hard to establish an accurate mathematical model of an ARV. An ARV’s dynamic system is quite complex due to the high nonlinearity, strong coupling, and hydrodynamic coefficients uncertainty. In addition, it is susceptible to unknown disturbances, such as sea currents and waves, resulting in oscillations and overshoots during control. (2) The working mode of the Haidou-1 ARV results in its various residual buoyancy states. The type, weight, and placement of the samples taken by the ARV varies greatly during each sampling session. This leads to changes in the physical parameters, which in turn leads to great changes in the hull hydrodynamic coefficient, affecting the stability and maneuverability of the ARV. (3) In order to conserve energy, extend operation time, and save space for more samples, the Haidou-1 ARV employs rotating propellers and under-actuated control, which undoubtedly increases the difficulty of control.

Therefore, the model-based control (MBC) methods (relying on accurate mathematical models), such as feedback linearization control and adaptive control, are difficult to use for a Haidou-1 ARV motion control system. It is difficult to ensure the stability and robustness of the control system without an accurate mathematical model. To address the above problems, we need a data-driven control method to address the Haidou-1 ARV motion control problem, which does not require any mathematical modeling information of an ARV.

Data-driven control methods are defined as the controller being designed with the direct use of the I/O data of the controlled system and the knowledge gained through data processing but no explicit mathematical model information of the controlled system [11], e.g., PID control [12], model-free adaptive control (MFAC) [13], correlation-based tuning control (CbT) [14], lazy learning control (LL) [15], iterative feedback tuning control (IFT) [16], etc. The PID control method is the simplest and most commonly used data-driven control method [17]. However, as the parameters of the controlled system change, the parameters of the PID controller should also be readjusted, and the adjustment of control parameters is more dependent on engineering experience. In addition, the PID method is more commonly applied to linear systems [18]. It is not suitable for the Haidou-1 ARV control system, which has high nonlinearity, time-varying, and structural changes.

The MFAC [19] method is an effective data-driven control method for nonlinear control systems. The MFAC method is perhaps the best solution to Haidou-1 ARV motion control in that: (1) It does not require any mathematical model information about the ARV, which is crucial for Haidou-1 ARV motion control, whose mathematical model is very hard to establish. (2) The MFAC method does not require any external test signals; it is simple but has a strong robustness [19]. (3) A number of MFAC applications have been reported in many fields, but few applications in robotics [20,21,22,23].

However, the conventional MFAC method suffers from drawbacks such as overshoot, oscillation, and slow convergence, which are caused by inevitable dynamic linearization error. To address the problem, we need to improve the adaptation capabilities of MFAC method. The sliding-mode control (SMC) [24,25] is perhaps the best solution because of the following: (1) Since SMC designs the system’s sliding-mode based on the desired dynamic characteristics of the system instead of system parameters and disturbances, the SMC has the features of insensitivity to parameter variations, fast response, simple physical implementation, etc. [26]. (2) SMC has been successfully applied in multiple different fields [27,28,29,30]. Therefore, SMC is an ideal algorithm that can improve the performance of MFAC method.

Drawing upon the aforementioned research findings, a data-driven, model-free adaptive sliding mode control (MFASMC) approach is proposed in this paper to address the Haidou-1 ARV motion control problem. The specific contributions of this paper are mainly in three aspects:Compared with the traditional MFAC methods used in [20,21,22,23], this paper further incorporates a discrete sliding mode control (SMC) method to enhance the system’s ability to withstand parameter uncertainties and external disturbances. The SMC method also guarantees MFAC method’s fast convergence and low overshooting.This paper analyzes two major difficulties in ARV motion control, i.e., various mathematical model states and the strong coupling effect in depth control. The method in this paper does not rely on any mathematical modeling information and is a data-driven control method. Even if the dynamic parameters of the ARV vary over a wide range and there are external disturbances, the proposed control method can maneuver the Haidou-1 ARV cruising at the desired forward speed, heading, and depth. Simulation outcomes of the proposed approach are contrasted with those of the PID control and MFAC control method. Compared results show that the proposed MFASMC method has better tracking performance for the ARV motion control system in the presence of disturbances.The control method in this paper can be implemented not only on ARVs but can also be extended to AUVs and other types of underwater robotic vehicles.

The rest of this paper is structured as follows: Section 2 builds the dynamical model of Haidou-1 ARV and analyzes two main difficulties in motion control of Haidou-1 ARV. Section 3 designs the MFASMC method. Section 4 provides the simulation studies. Conclusions are given in Section 5.

## 2. Dynamical Model and Analysis of Haidou-1 ARV Motion Control

The main physical information of the Haidou-1 ARV is shown in Table 1.

The arrangement of Haidou-1 ARV motion control system is shown in Figure 2. Two main thrusters are fixed on the rotating elevators at the stern. The elevators are driven by rudder motors and can be controlled independently, with a rotation angle of ±90 degrees. Two fixed vertical thrusters are located at the front.

### 2.1. Kinematics

As shown in Figure 3, we define two coordinate frames named North-East-Down coordinate system and body-fixed frame [31,32]. We set the origin of the body-fixed frame at Haidou-1 ARV’s center of buoyancy. The main kinematics parameters of Haidou-1 ARV in six degrees of freedom are shown in Table 2.

We have following vectors:(1)$$\eta_{1}=\left[\begin{array}{c} x\\ y\\ z \end{array}\right],\;\eta_{2}=\left[\begin{array}{c} \phi \\ \theta \\ \psi \end{array}\right],\;\eta =\left[\begin{array}{c} \eta_{1}\\ \eta_{2} \end{array}\right]\;V_{1}=\left[\begin{array}{c} u\\ v\\ w \end{array}\right],\;V_{2}=\left[\begin{array}{c} p\\ q\\ r \end{array}\right],\;V=\left[\begin{array}{c} V_{1}\\ V_{2} \end{array}\right]\;F_{1}=\left[\begin{array}{c} X\\ Y\\ Z \end{array}\right],\;F_{2}=\left[\begin{array}{c} K\\ M\\ N \end{array}\right],\;F=\left[\begin{array}{c} F_{1}\\ F_{2} \end{array}\right]$$
where η is the position and Euler angle vector, V is the linear and angular velocities vector, and F is the forces and moment vector.

We have the kinematics model as follows:(2)$$\left[\begin{array}{c} \eta_{1}\\ \eta_{2} \end{array}\right]=\left[\begin{array}{cc} R_{1}(\eta_{2}) & 0_{3\times 3}\\ 0_{3\times 3} & R_{2}(\eta_{2}) \end{array}\right]$$
in which
(3)$$R_{1}(\eta_{2})=\left[\begin{array}{ccc} c\psi c\theta & c\psi s\theta s\phi -s\psi c\phi & s\psi s\phi +c\psi c\phi s\theta \\ s\psi c\theta & c\psi c\phi +s\psi s\theta s\phi & s\theta s\psi c\phi -c\psi s\phi \\ -s\theta & c\theta s\phi & c\theta c\phi \end{array}\right]\;R_{2}(\eta_{2})=\left[\begin{array}{ccc} 1 & s\phi t\theta & c\phi t\theta \\ 0 & c\phi & -s\phi \\ 0 & s\phi /c\theta & c\phi /c\theta \end{array}\right]$$
And in which s=sin, c=cos, and t=tan.

### 2.2. Dynamics

The six-degree-of-freedom dynamics model of the Haidou-1 ARV is as follows [32]:(4)$$M_{T}=-C(V)V-D(V)V-G(\eta_{2})+F_{thrust}+F_{rudder}+F_{env}$$
where $M_{T}$ is the total mass matrix, C(V) is the Coriolis–centripetal matrix, D(V) is the hydrodynamic damping matrix, $G(\eta_{2})$ is the restoring force matrix, $F_{thrust}$ are the thruster forces, $F_{rudder}$ are the rudder forces, $F_{env}$ are the environmental forces including wave forces and wind forces, and $V={\left[\begin{array}{ccc} u & v & \begin{array}{ccc} w & p & \begin{array}{cc} q & r \end{array} \end{array} \end{array}\right]}^{T}$ is the linear and angular velocities vector.

$F_{thrust}$ can be expressed as the following equation:(5)$$F_{thrust}=\left[\begin{array}{c} F_{sLx}+F_{sRx}\\ 0\\ -F_{sLy}-F_{sRy}+F_{bL}+F_{bR}\\ 0\\ F_{sLx}*zstn+F_{sRx}*zstn+F_{sLy}*xstn+F_{sRy}*xstn-F_{bL}*xbow-F_{bR}*xbow\\ F_{sLx}*ystn-F_{sRx}*ystn \end{array}\right]$$
where $F_{sLx}=F_{sL}*\cos(dsL)$ is the projection of left main thruster force $F_{sL}$ in the x direction, dsL is the angle of left elevator, $F_{sRx}=F_{sR}*\cos(dsR)$ is the projection of right main thruster force $F_{sR}$ in the x direction, dsR is the angle of right elevator, $F_{sLy}=F_{sL}*\sin(dsL)$ is the projection of left main thruster force $F_{sL}$ in the y direction, $F_{sRy}=F_{sR}*\sin(dsR)$ is the projection of right main thruster force $F_{sR}$ in the y direction, xstn, ystn, and zstn are the coordinates of the main thruster in the body-fixed frame, and xbow is the x-direction coordinates of the vertical thruster in the body-fixed frame.

TsLMatrix can be expressed as the following:(6)$$TsLMatrix=\left[\begin{array}{ccc} JsL^{2} & JsL & 1 \end{array}\right]$$

JsL is the advance coefficient and can be expressed as the following:(7)JsL=u∗cos(dsL)/(D∗nsL)

where u is the forward speed of the ARV, D is the diameter of the thruster, and nsL is the speed of the left main thruster.

The right main thruster force matrix TsRMatrix is similar to the TsLMatrix. The TbLMatrix can be expressed as follows:(8)$$TbLMatrix=\left[\begin{array}{ccc} nbL^{3} & nbL^{2} & nbL \end{array}\right]$$
where nbL is the speed of the left vertical thruster.

The right vertical thruster force matrix TbRMatrix is similar to the TbLMatrix.

$F_{rudder}$ can be expressed as follows:(9)$$F_{rudder}=rdMatrix*rdCoeffMatrix$$
where rdMatrix is the rudder force matrix, which can be expressed as follows:(10)$$rdMatrix=\left[\begin{array}{cccc} u^{2}dsR^{2} & u^{2}dsL^{2} & u^{2}dsR & u^{2}dsL \end{array}\right]$$

rdCoeffMatrix is the rudder force coefficient matrix which can be obtained by a CFD simulation.

### 2.3. Analysis of Haidou-1 ARV Motion Control

The principle of Haidou-1 ARV motion control system is shown in Figure 4. Haidou-1 ARV can cruise at a desired depth, heading, and forward speed near the seabed. We can decouple the ARV motion into a horizontal plane and vertical plane. In the horizontal plane we need to control the heading and forward speed, in the vertical plane we need to control the depth. The heading control and forward speed control are realized by the two main thrusters cooperating differentially, as shown in Figure 5. In Figure 4, the output of heading and forward speed controller in horizontal plane is the desired propeller speed.

Currently, the vertical plane motion control of the Haidou-1 ARV relies on twin fixed vertical thrusters located at the front and two rotated elevators with thrusters fixed on, as shown in Figure 6a. The advantage of this method is its simplicity of implementation. However, the use of vertical thrusters consumes more energy. There are two reasons for this phenomenon: Firstly, the direction of the incoming current is perpendicular to the direction of the vertical thrusters, which results in a lower efficiency of the vertical propellers compared to that of the open-water propeller. Secondly, because the vertical thrusters are tightly attached to the body of Haidou-1 ARV, the rapid rotation of the vertical propellers can cause a great change in the flow field near Haidou-1 ARV, which generates a larger pressure drag, as well as sailing drag, so it will consume more energy for the main thrusters to maintain the ARV’s constant depth and constant forward speed. For the above reasons, this paper mainly focuses on the study of the Haidou-1 ARV, which only uses elevators and rotated main thrusters at the stern to finish under-actuated motion control, as shown in Figure 6b. As shown in Equation 5, we set TbLMatrix and TbRMatrix to zero. After simulation by CFD, we found that this method can reduce propulsion system energy consumption by 58.3% compared with using the vertical thrusters method.

There are two main difficulties in the motion control of Haidou-1 ARV.

The working mode of ARV leads to its various mathematical model states. Haidou-1 ARV has the ability of grabbing up to 30 kg samples. When an ARV grasps objects or takes sediment samples with a manipulator, the collected samples will cause changes in the center of gravity, metacentric height, hydrostatics parameters and residual buoyancy of the ARV, which will lead to corresponding changes in the mathematical model of the ARV and can be seen as a disturbance to ARV motion control. In Figure 3, when the center of buoyancy of Haidou-1 ARV is set to (0, 0, 0), then the coordinates of its center of gravity are 0 mm, 0 mm, and 43.95 mm. When Haidou-1 ARV grasps a 30 kg sample, the coordinates of center of buoyancy change to 5.2 mm, 0.9 mm, and 2.1 mm, and the coordinates of its center of gravity change to 15.5 mm, 2.5 mm, and 38.05 mm. When the metacentric height is reduced by 8 mm, the great change in the metacentric height will greatly increase the difficulty of control.There is coupling between Haidou-1 ARV’s six degrees of freedom. The coupling effect is stronger when we only use rotated thrusters to realize depth control. Because we only use elevators and rotated main thrusters to accomplish depth control, when there is a deviation between the actual depth and the required depth, we need to adjust the elevator angle and main thruster speed to approach the desired depth, which will inevitably cause a change in the forward speed of an ARV. The change in forward speed leads to changes in the surrounding flow velocity, which in turn causes variations in the forces acting on the elevators and changes in the forces of thrusters (Equation (9)). In order to achieve simultaneous depth control and forward speed control, it is inevitable to frequently adjust the angle of the elevators and the speed of the thrusters over a long period of time. This may lead to significant overshooting and even cause oscillations or failure to converge, resulting in instability in ARV motion control.

Considering the difficulties in the motion control of the Haidou-1 ARV mentioned above, we next discuss the data-driven MFASMC method.

## 3. MFASMC Controller Design

In order to design MFASMC controller, first we need to design MFAC controller. The underlying idea of MFAC method is that we use an equivalent dynamic linearization data model with a novel concept called pseudo-partial derivative (PPD) at each operation point to replace the discrete-time nonlinear system; then we estimate the PPD by only using the I/O information; and finally, we design the one-step-forward controller which is called the MFAC controller [11]. There are three dynamic linearization data models. Reference [33] converted the unmanned surface vehicle (USV) input and output data into a compact form dynamic linearization (CFDL) model. Reference [34] proposed the model-free adaptive fault-tolerant control based on a partial form dynamic linearization (PFDL) model for high-speed trains with traction/braking force constraints and actuator faults. However, the CFDL and PFDL method only considers the relationship between the output change of the system at the next moment and the input change at the current moment. However, the output signal of the ARV motion control system depends not only on the control input at a certain moment. In this paper, we use the full form dynamic linearization (FFDL) method to investigate the ARV motion control problem. In the FFDL method, the influence of the input signal and output signal in a fixed length sliding time on the output signal at the next moment can be taken into account when the data is linearized.

Section 3 is structured as follows: In Section 3.1 we change the ARV motion control system to an equivalent dynamic linearization data model using the FFDL method. In Section 3.2 the FFDL-MFAC scheme design is given based on the FFDL data model in Section 3.1. In Section 3.3, we will introduce the SMC method to form the MFASMC control method.

### 3.1. Full-Form Dynamic Linearization Method

A SISO discrete-time nonlinear system can be expressed as follows:(11)$$y(k+1)=f(y(k),\ldots ,y(k-n_{y}),u(k),\ldots ,u(k-n_{u}))$$
where y(k) and u(k), respectively, are the system output and input at time k, $n_{y}$ and $n_{u}$ the unknown orders, and f(…) is an unknown nonlinear function. Haidou-1 ARV motion system can be expressed by Equation (11).

We define $H_{L_{y},L_{u}}(k)\in R^{L_{y}+L_{u}}$, then we have
(12)$$H_{L_{y},L_{u}}(k)={\left[y(k),\cdot \cdot \cdot ,y(k-L_{y}+1),u(k),\cdot \cdot \cdot ,u(k-L_{u}+1)\right]}^{T}$$
where k≤0, $H_{L_{y},L_{u}}(k)=0_{L_{y},L_{u}}$, and where $L_{y}$ and $L_{u}$ ($0\leq L_{y}\leq n_{y},0\leq L_{u}\leq n_{u}$) are the pseudo-order of the system output and input, respectively.

Before introducing FFDL, we make the following assumptions [31]:

**Assumption** **1.***The partial derivatives of* f(…)*with respect to every variable are continuous*.

**Assumption** **2.***System (11) is generalized Lipschitz, that is,* $\left|y(k_{1}+1)-y(k_{2}+1)\right|\leq b\left\|H_{L_{y},L_{u}}(k_{1})-H_{L_{y},L_{u}}(k_{2})\right\|$*, where* $y(k_{i}+1)=f(y(k_{i}),\ldots ,y(k_{i}-n_{y}),u(k_{i}),\ldots ,u(k_{i}-n_{u}))$*, and* i=1,2*,* b>0.

From a practical point of view, the above assumptions are reasonable and acceptable. Assumption 1 is a typical constraint on general nonlinear systems in control system design. Assumption 2 places a limitation on the rate of change of the controller output. The Haidou-1 ARV motion system can meet these assumptions.

For the nonlinear system (11), satisfying Assumption 1 and 2, for $L_{y}$ and $L_{u}$, there exists a parameter vector, $\phi (k)\in R^{L_{y}+L_{u}}$, called the pseudo-gradient (PG), such that system (11) can be transformed into the following equivalent FFDL description: (13)Δy(k+1)=ϕ(k)ΔH(k)
where for any k, $\phi (k)=\left[\phi_{1}(k),\cdot \cdot \cdot ,\phi_{L_{y}}(k),\phi_{L_{y}+1}(k)\cdot \cdot \cdot \phi_{L_{y}+L_{u}}(k)\right]$ is bounded.

### 3.2. FFDL-MFAC Method

Consider the following control input criterion function:(14)$$J(u(k))={\left|y^{*}(k+1)-y(k+1)\right|}^{2}+\lambda {\left|u(k)-u(k-1)\right|}^{2}$$
where $y^{*}(k+1)$ is the desired output signal, and λ>0 is a weighting factor.

Substituting (13) into (14), differentiating with respect to u(k), and letting it zero, we obtain the following:(15)$$\begin{array}{l} u(k)=u(k-1)+\frac{\rho_{L_{y}+1}\phi_{L_{y}+1}(k)(y^{*}(k+1)-y(k))}{\lambda +{\left|\phi_{L_{y}+1}(k)\right|}^{2}}\\ -\frac{\phi_{L_{y}+1}(k)(\sum_{i=1}^{L_{y}}\rho_{i}\phi_{i}(k)\Delta y(k-i+1)+\sum_{i=L_{y}+2}^{L_{y}+L_{u}}\rho_{i}\phi_{i}(k)\Delta u(k-L_{y}-i+1))}{\lambda +{\left|\phi_{L_{y}+1}(k)\right|}^{2}} \end{array}$$

We define the index function for the PG estimation as follows:(16)$$J(\phi (k))={\left|y(k)-y(k-1)-\phi^{T}(k)\Delta H(k-1)\right|}^{2}+\mu {\left\|\phi (k)-\overset{\wedge }{\phi }(k-1)\right\|}^{2}$$
where μ>0 is a weighting factor, and $\overset{\wedge }{\phi }(k)$ is the estimation value of ϕ(k).

Differentiating (16) with respect to ${\overset{\wedge }{\phi }}_{f,L_{y},L_{u}}(k)$, and letting it zero, we obtain the following:(17)$$\begin{array}{cc} \overset{\wedge }{\phi }(k) & =\overset{\wedge }{\phi }(k-1)+\\ & \frac{\eta \Delta H(k-1)(\Delta y(k)-\overset{\wedge }{\phi }(k-1)\Delta H(k-1))}{\mu +{\left\|\Delta H(k-1)\right\|}^{2}} \end{array}$$
(18)$$\overset{\wedge }{\phi }(k)=\overset{\wedge }{\phi }(1)\;\mathrm{if}\;\left\|\overset{\wedge }{\phi }(k)\right\|\leq \varepsilon \;\mathrm{or}\;\left\|\Delta H(k-1)\right\|\leq \varepsilon \;\mathrm{or}\;sign({\overset{\wedge }{\phi }}_{L_{y}+1}(k))\neq sign({\overset{\wedge }{\phi }}_{L_{y}+1}(1))$$
where η∈(0,2] is a step-size constant, μ>0 is called the weight coefficient, and $\overset{\wedge }{\phi }(1)$ is the initial value of $\overset{\wedge }{\phi }(k)$.

Equation (17) is the PG estimation algorithm. Equation (18) is the PG reset algorithm.

Based on Equations (15) and (17), the FFDL-MFAC [35] approach can be expressed as the following:(19)$$\begin{array}{cc} u_{MFA}(k) & =u_{MFA}(k-1)+\\ & \frac{\rho_{L_{y}+1}(k){\overset{\wedge }{\phi }}_{L_{y}+1}(k)(y^{*}(k+1)-y(k))}{\lambda +{\left|{\overset{\wedge }{\phi }}_{L_{y}+1}(k)\right|}^{2}}\\ & -\frac{{\overset{\wedge }{\phi }}_{L_{y}+1}(k)(\sum_{i=1}^{L_{y}}\rho_{i}\overset{\wedge }{\phi_{i}}(k)\Delta y(k-i+1)+\sum_{i=L_{y}+2}^{L_{y}+L_{u}}\rho_{i}\overset{\wedge }{\phi_{i}}(k)\Delta u(k-L_{y}-i+1))}{\lambda +{\left|{\overset{\wedge }{\phi }}_{L_{y}+1}(k)\right|}^{2}} \end{array}$$
where λ>0 is called the weight factor, $\rho_{i}\in \left(0,1\right]$ is a step-size constant and $i=1,2,\cdot \cdot \cdot ,L_{y}+L_{u}$, ε is a sufficiently small positive number. 

Equation (19) is the FFDL-MFAC method control law. 

The FFDL-MFAC method only uses the I/O data to estimate the PG value; the estimated PG value and the one-step-forward error are substituted into the control law. The control process is shown in Figure 7.

### 3.3. MFASMC Method Design

The output of MFAC method is donated as $y_{MFA}(k)$.

For the model (13), we set the discrete sliding surface as the following:(20)s(k)=e(k)

so,
(21)s(k+1)=e(k+1)

We can transform Equation (13) into the following:(22)$$\begin{array}{c} \Delta y(k+1)=\sum_{i=1}^{L_{y}}\overset{\wedge }{\phi_{i}}(k)\Delta y(k-i+1)\\ +\sum_{i=L_{y}+2}^{L_{y}+L_{u}}\overset{\wedge }{\phi_{i}}(k)\Delta u(k+L_{y}-i+1)+{\overset{\wedge }{\phi }}_{L_{y}+1}(k)\Delta u(k) \end{array}$$

The reaching law [36] is given as follows:(23)$$s(k+1)=s(k)-q_{1}h\cdot s(k)-q_{2}h\cdot sig^{\alpha }(s(k))$$
where $sig^{\alpha }(s(k))=\mathrm{sgn}(s(k))\cdot {\left|s(k)\right|}^{\alpha }$ and $0<q_{1}h<1$, $0<q_{2}h<1$, 0<α<1. T is the discrete-time sample period. Substituting Equations (21) and (22) into Equation (23), gives the following:(24)$$\begin{array}{l} \Delta u_{SMC}(k)=\frac{1}{{\overset{\wedge }{\phi }}_{L_{y}+1}(k)}[(1-q_{1}h)s(k)\\ -q_{2}hsig^{\alpha }(s(k))-\sum_{i=1}^{L_{y}}\overset{\wedge }{\phi_{i}}(k)\Delta y(k-i+1)\\ -\sum_{i=L_{y}+2}^{L_{y}+L_{u}}\overset{\wedge }{\phi_{i}}(k)\Delta u(k+L_{y}-i+1)] \end{array}$$

The final MFASMC control law is Equation (19) plus Equation (24).
(25)$$u(k)=u_{MFA}(k)+u_{SMC}(k)$$

A MFASMC motion control block diagram is illustrated in Figure 8.

## 4. Simulation

To validate the effectiveness of the MFASMC controller proposed in this paper for Haidou-1 ARV motion control, simulation experiments were designed for multiple scenarios. The simulation model adopts the Haidou-1 ARV six-degree-of-freedom hydrodynamic model in [37].

### 4.1. Horizontal Plane Motion Simulation

The horizontal plane motion includes directional and constant speed navigation. The experimental scenario we designed is that the ARV performs stepwise steering maneuvers of ±20° and ±90° at a speed of 1 knot, corresponding to small-scale and large-scale steering movements, respectively. The time interval for a 20° heading step is 35 s, and the time interval for a 90° heading step is 20 s. The control cycle is selected as 0.5 s, consistent with the actual control cycle of the ARV. The pseudo-order selection for MFASMC is $L_{y}=2$ and $L_{u}=1$. The main control parameters of heading control and speed control are shown in Table 3. The heading step response curve and forward speed response curve are shown in Figure 9. The performance index of heading controller step response is shown in Table 4.

From Figure 9 and Table 4, we note that all three methods can achieve the control effect under undisturbed conditions. MFASMC and MFAC exhibit similar response performance, MFASMC has a larger overshoot in the 20° step steering maneuver but has a shorter adjusting time in the 90° step steering maneuver. In contrast, under the action of PID, a larger overshoot and longer adjusting time appear.

In order to further analyze the performance of the three controllers, we consider the conditions of the hydrodynamic coefficients change and environmental disturbances. For example, ARV replaced the installed payload equipment and constant disturbances caused by sea currents. We add a constant disturbance of 20 Nm in the yaw direction. In addition, we choose three hydrodynamic coefficients that have the most significant impact on control effectiveness $I_{z}$, $N_{r}$, and $N_{r\left|r\right|}$, reducing the values by 80%. The control performances of the three methods are shown in Figure 10. The performance index of heading controller step response is shown in Table 5.

By analyzing the response curves in Figure 10 and the performance indicators in Table 5, it can be observed that both PID and MFAC exhibit significant overshoot, and the adjustment time is noticeably prolonged, especially the overshoot of PID reaches to 183.1%. It implies that PID does not show a good robustness and adaptability once external disturbances exist. In contrast, despite the constant disturbance and hydrodynamic coefficients change, MFASMC still maintains a consistent control performance. MFASMC maintains less than 2% overshoot and a shorter adjusting time.

From simulation results above, we can find that compared with PID method and MFAC method, the MFASMC control method for ARV heading and forward speed control has better control performance, though the dynamic parameters of an ARV vary over a wide range and external disturbances exist. The proposed MFASMC control method has strong robustness and adaptability.

### 4.2. Vertical Plane Control Simulation

This section will show the simulations of Haidou-1 ARV under-actuated depth control using the three control methods. The experimental scenario we designed includes the Haidou-1 ARV operation with a stepwise steering and constant forward speed, as discussed in Section 4.1. At the beginning of this maneuver the Haidou-1 ARV was at the surface; when simulation starts, we set the command depth to 5 m, at t = 40 s, the command depth changes to 10 m. The Haidou-1 ARV only used rotating propellers to finish under-actuated depth control, this method can save energy, increase operational duration, and save space for placing more samples on the ARV.

The pseudo-order selection for MFASMC is $L_{y}=2$ and $L_{u}=1$. The main control parameters of depth control are set as Table 6. Figure 11 shows some simulation results obtained during depth changing maneuvers. Table 7 shows the performance index of the controller step response.

From Figure 11 and Table 7, we note that all three of the control methods can be implemented using only rotated propellers to achieve Haidou-1 ARV depth control under undisturbed conditions. MFASMC and MFAC exhibit a similar response performance. Compared with PID, MFASMC and MFAC have smaller overshoots but longer adjusting time.

In order to further analyze the performance of the three controllers, consider the condition of the hydrodynamic coefficients change and environmental disturbances. For example, the ARV finishes sampling and restarts an autonomous search. We add a constant disturbance of 30 kg in the heave direction and reduce the metacentric height of the ARV by 20%. The control performances of the three methods are shown in Figure 12. The performance index of the depth controller step response is shown in Table 8.

By analyzing the response curves in Figure 12 and the performance indicators in Table 8, it can be observed that due to the influence of the model parameter mismatch, the PID exhibits significant oscillations, and the overshoot reaches to 37.4%. In contrast, despite the constant disturbance, both the MFAC and MFASMC can achieve the control effect under disturbed conditions. But MFAC exhibits larger overshoot up to 62% and longer adjusting time. MFASMC still maintains a consistent control performance. Compared with the undisturbed condition, the overshoot and adjusting time of MFASMC experience little change with the stochastic model parameter perturbation and environmental disturbances.

From the simulation results above, we find that for the Haidou-1 ARV under-actuated depth control problem, compared with PID method and MFAC method, the MFASMC method achieves a better performance, though the dynamic parameters of an ARV vary over a wide range and external disturbances. The proposed MFASMC control method has a strong robustness and adaptability.

## 5. Conclusions

In this study, we propose a data-driven, model-free adaptive sliding mode control (MFASMC) approach to address the Haidou-1 ARV motion control problem with uncertainties. The following conclusions can be obtained through simulation results:The theoretical analysis of the Haidou-1 ARV motion control problem indicates that it will reduce propulsion system energy consumption greatly if we only use the elevators and the rotated main thrusters at the stern to finish under-actuated motion control. The difficulties of the ARV motion control can be summarized as a great hydrodynamic coefficients change and a strong coupling effect.To address the above problem, we propose the MFASMC method. We theoretically analyzed the working principles of this method. The MFAC method relies only on the real-time measurement data of an ARV instead of any mathematical modeling information, and the SMC method guarantees the MFAC method’s fast convergence and low overshooting. It is an ideal method to deal with Haidou-1 ARV’s motion control problem.The simulation results show that compared with PID and MFAC methods, though the hydrodynamic coefficients change and there are environmental disturbances, MFASMC still has good control performance and shows features of strong robustness and adaptability. It can also achieve the goal that Haidou-1 ARV under-actuated motion control only use the elevators and main thrusters, so it is feasible and effective for Haidou-1 ARV motion control. Haidou-1 ARV can cruise at the desired forward speed, heading, and depth even when the dynamic parameters of the ARV vary over a wide range and external disturbances exist.

## Figures and Tables

**Figure 1 sensors-24-03592-f001:**
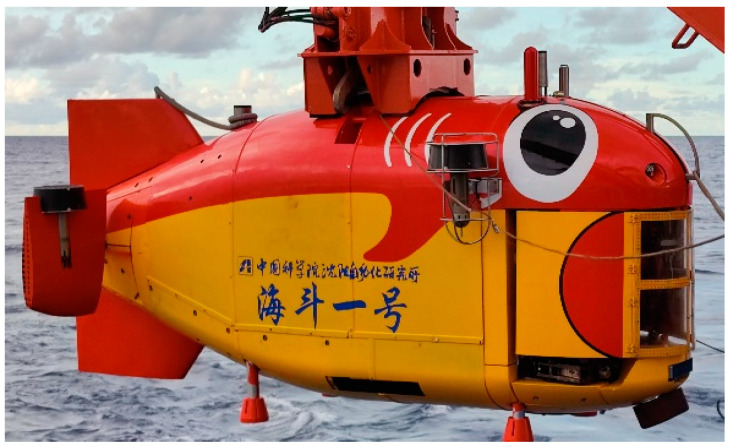
Haidou-1 ARV. The meaning of Chinese character in this figure is Shenyang Institute of Automation, CAS, Haidou-1.

**Figure 2 sensors-24-03592-f002:**
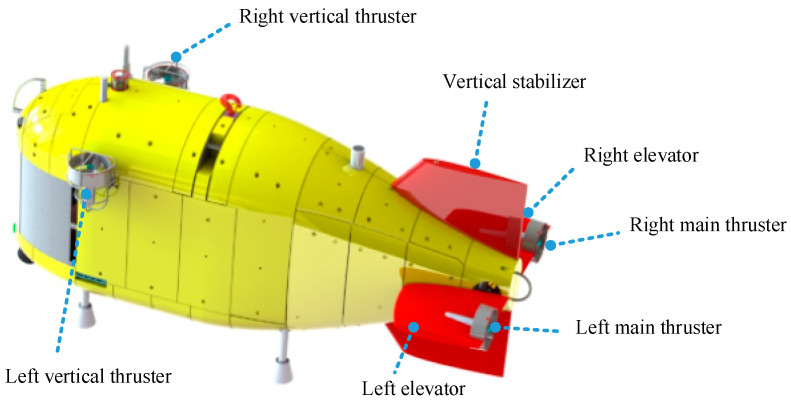
The arrangement of Haidou-1 ARV motion control system.

**Figure 3 sensors-24-03592-f003:**
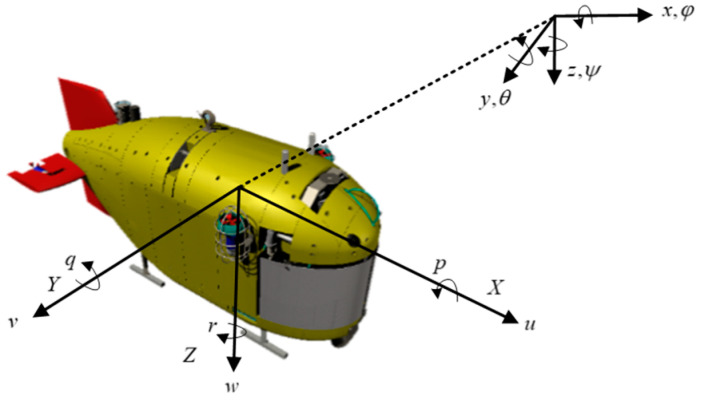
Haidou-1 ARV dynamics in different coordinates.

**Figure 4 sensors-24-03592-f004:**
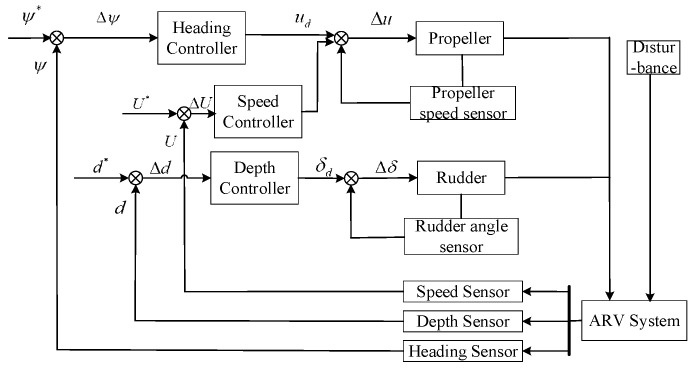
The principle of the Haidou-1 ARV motion control system. the character * refers to the expected output signal.

**Figure 5 sensors-24-03592-f005:**
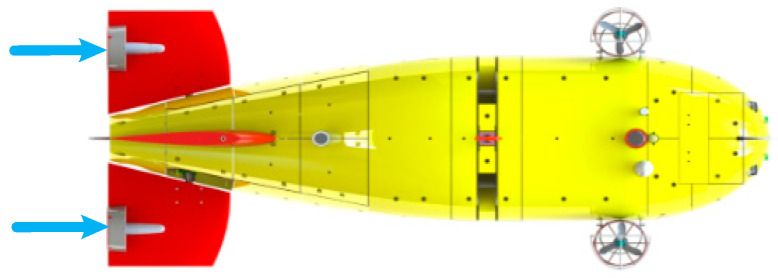
The motion control of Haidou-1 ARV in the horizontal plane. the blue arrows refers to the thrust direction of main thrusters.

**Figure 6 sensors-24-03592-f006:**
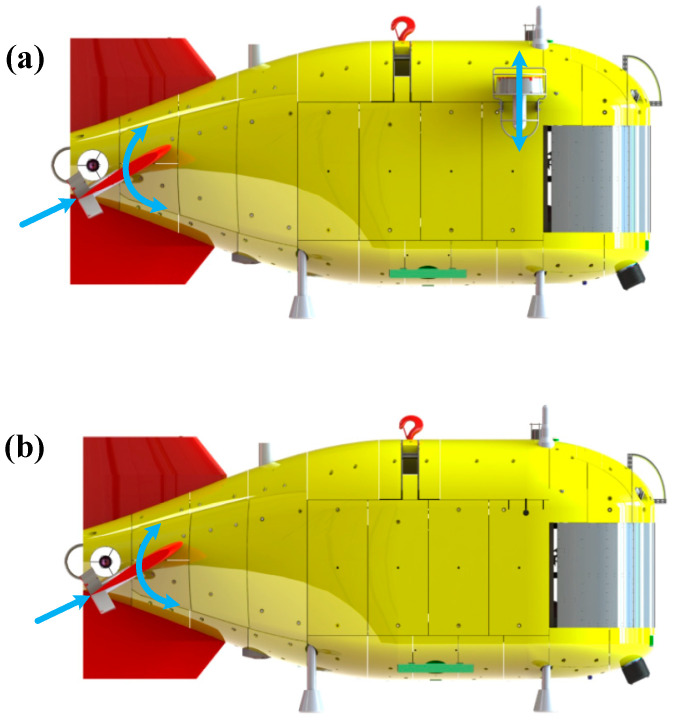
Motion control of Haidou-1 ARV in vertical plane: (**a**) is the thrust allocation diagram of full-actuated depth control; (**b**) is the thrust allocation diagram of under-actuated depth control. The blue arrows refers to the thrust direction of main thrusters and the direction of elevators rotation.

**Figure 7 sensors-24-03592-f007:**
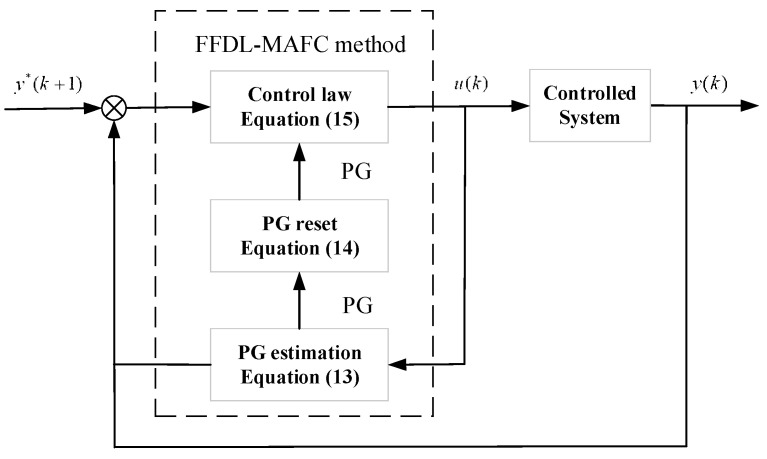
FFDL-MFAC control structure.

**Figure 8 sensors-24-03592-f008:**
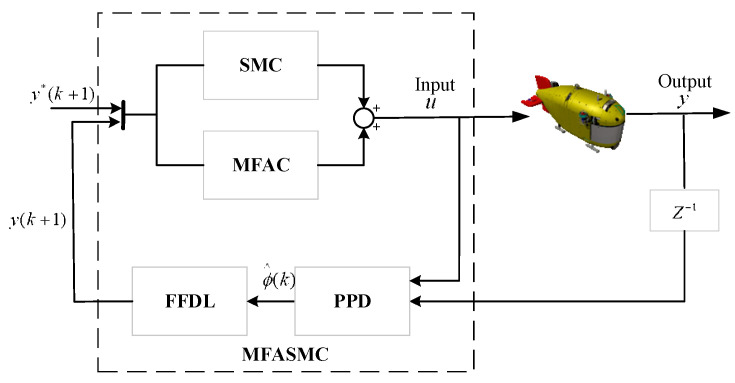
A MFASMC motion control block diagram.

**Figure 9 sensors-24-03592-f009:**
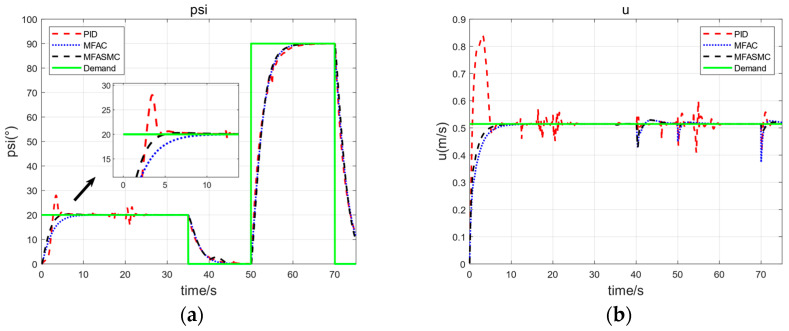
Horizontal plane motion simulation result (without disturbance and hydrodynamic coefficients change). (**a**) Heading step response curve and (**b**) forward speed response curve. MFASMC control performance (black) compared with both PID (red) and MFAC (blue) control performance.

**Figure 10 sensors-24-03592-f010:**
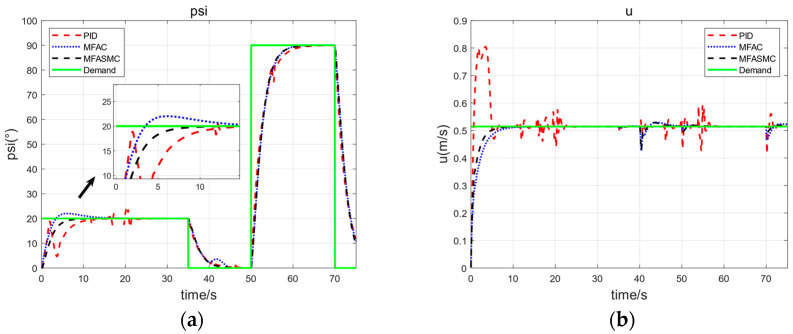
Horizontal plane motion simulation result (with 20 Nm constant disturbance and hydrodynamic coefficients change). (**a**) Heading step response curve and (**b**) forward speed response curve. MFASMC control performance (black) compared with both PID (red) and MFAC (blue) control performance.

**Figure 11 sensors-24-03592-f011:**
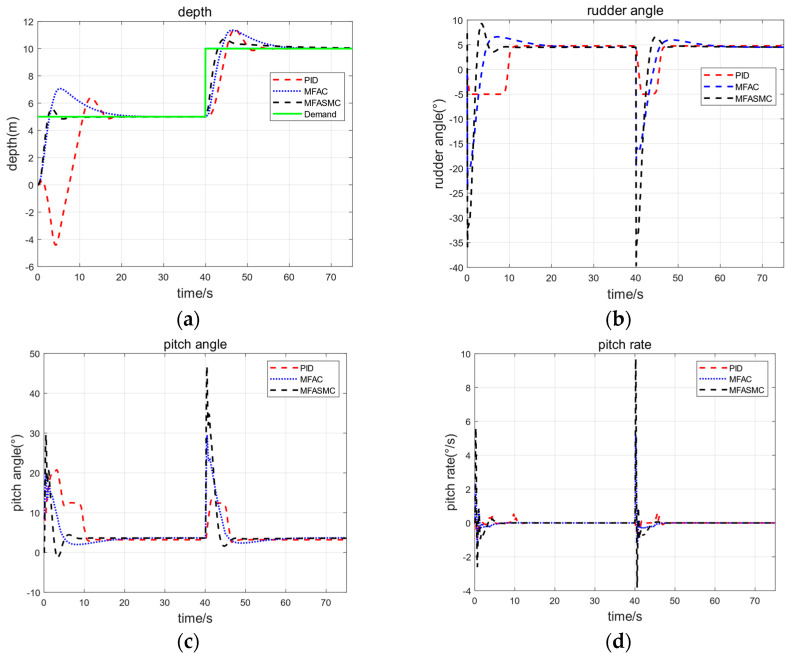
Simulation results in vertical plane (without disturbance). (**a**) is depth step response curve and (**b**–**d**) are the rudder angle, pitch angle and pitch rate curves respectively. MFASMC control performance (black) compared with both PID (red) and MFAC (blue) control performance.

**Figure 12 sensors-24-03592-f012:**
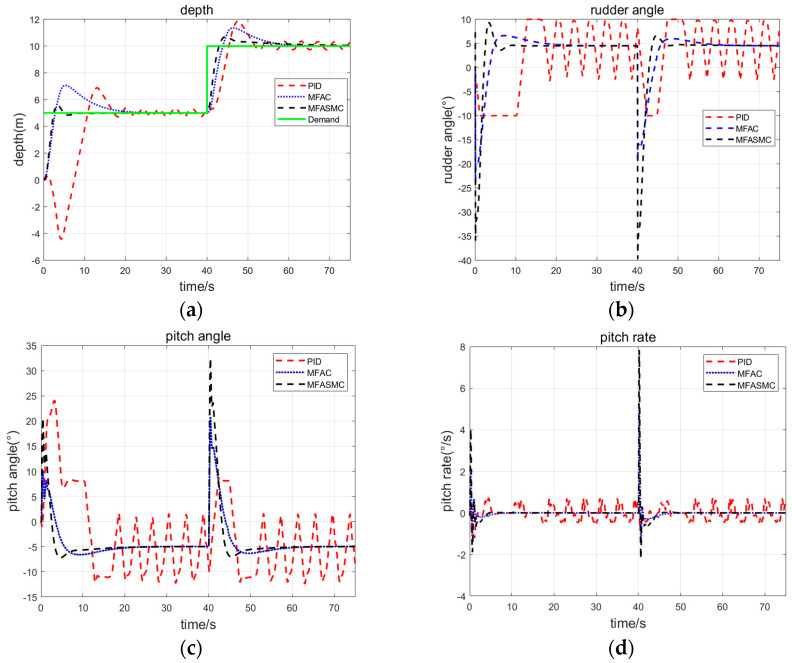
Simulation result (with 30 kg constant disturbance and hydrodynamic coefficients change). (**a**) Depth step response curve and (**b**–**d**) are the rudder angle, pitch angle and pitch rate curves, respectively. MFASMC control performance (black) compared with both PID (red) and MFAC (blue) control performance.

**Table 1 sensors-24-03592-t001:** Main physical information of Haidou-1 ARV.

Parameter	Value
Dimension	3.8 × 1.1× 1.6 m (L × W × H)
Weight	2640 kg in air
Depth of operation	Maximum 11,000 m
Time of operation	10 h formal operation and with reserve
Propulsion	Two rotated main thrusters and two vertical thrusters
Center of buoyancy	[0.0, 0.0, 0.0] m
Center of gravity	[0.0, 0.0, 0.044] m
Coordinates of main thruster	[−1.72, 0.6, −0.02] m
Coordinates of vertical Thruster (x-direction)	0.75 m

**Table 2 sensors-24-03592-t002:** Main kinematics parameters of Haidou-1 ARV.

DOF	Forces/Moment	Velocities	Positions/Euler Angles
Surge	X	u	x
Sway	Y	v	y
Heave	Z	w	z
Roll	K	p	ϕ
Pitch	M	q	θ
Yaw	N	r	ψ

**Table 3 sensors-24-03592-t003:** The main control parameters of MFASMC in horizontal plane motion.

Motion	Control Parameters	Value
Heading control	$\lambda_{h}$	0.5
	$\rho_{h}$	${\left[\begin{array}{ccc} 0.7 & 0.7 & 0.7 \end{array}\right]}^{T}$
	$\mu_{h}$	1
	$\eta_{h}$	0.2
Speed control	$\lambda_{s}$	0.001
	$\rho_{s}$	${\left[\begin{array}{ccc} 0.7 & 0.7 & 0.7 \end{array}\right]}^{T}$
	$\mu_{s}$	1
	$\eta_{s}$	0.2

**Table 4 sensors-24-03592-t004:** Performance index of heading controller step response (without disturbance and hydrodynamic coefficients change).

Step Amplitude	Controller	Overshoot/%	Rise Time/s	Adjusting Time/s
20°	PID	26.2	3.4	16.3
	MFAC	0	15.2	15.2
	MFASMC	2.9	7.8	15.5
90°	PID	0	18.7	18.7
	MFAC	0	16.3	16.3
	MFASMC	0	10.4	10.4

**Table 5 sensors-24-03592-t005:** Performance index of heading controller step response (with 20 Nm constant disturbance and hydrodynamic coefficients change).

Step Amplitude	Controller	Overshoot/%	Rise Time/s	Adjusting Time/s
20°	PID	183.1	/	27
	MFAC	30.2	3	24
	MFASMC	1.6	3.4	15.3
90°	PID	0	18	18
	MFAC	0	16.7	16.7
	MFASMC	0	10	10

**Table 6 sensors-24-03592-t006:** The main control parameters of MFASMC in vertical plane control.

Motion	Control Parameters	Value
Depth control	$\lambda_{d}$	0.5
	$\rho_{d}$	${\left[\begin{array}{ccc} 0.7 & 0.7 & 0.7 \end{array}\right]}^{T}$
	$\mu_{d}$	1
	$\eta_{d}$	0.2

**Table 7 sensors-24-03592-t007:** Performance index of depth controller step response (without disturbance and hydrodynamic coefficients change).

Step Amplitude	Controller	Overshoot/%	Rise Time/s	Adjusting Time/s
5 m	PID	18.2	14.2	24.6
	MFAC	12.9	7.5	24.8
	MFASMC	11.6	4.5	27.1
10 m	PID	8.9	9.8	21
	MFAC	12.4	7.5	26.8
	MFASMC	10.4	5.7	26.5

**Table 8 sensors-24-03592-t008:** Performance index of depth controller step response (with 30 kg constant disturbance and hydrodynamic coefficients change).

Step Amplitude	Controller	Overshoot/%	Rise Time/s	Adjusting Time/s
5 m	PID	37.4	13.2	/
	MFAC	62	5.2	25.3
	MFASMC	11.6	3.3	9
10 m	PID	18.8	7.6	/
	MFAC	11.7	7.2	25.4
	MFASMC	7	4.5	24.5

## Data Availability

The data are contained within the article.

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
