# Peer review of "Under-Actuated Motion Control of Haidou-1 ARV Using Data-Driven, Model-Free Adaptive Sliding Mode Control Method"

_sensors, 2024, doi:10.3390/s24113592_

Round 1

Reviewer 1 Report

Comments and Suggestions for Authors

The paper discusses the motion control of Autonomous and Remotely-operated Vehicle (ARV). The authors propose a data-driven model-free adaptive sliding mode control (MFASMC) to tackle the problem of motion control in Haidou-1 ARV. The proposed method was validated using a simulation.  

Here are some comments regarding the paper: 

1. I hope there are some results in real ARV. 

2. I have read the paper and still have not found the physical information of the used ARV. Moreover, if 1. is not possible, I would like to see the simulation results that involved the physical information as well as the environments. 

3. Please avoid using the formula or equation (Eq.) with a wordy expression, especially in Eq. (5).

4. Please use a vector style for figures.

Reviewer 2 Report

Comments and Suggestions for Authors

In this study, the authors propose a data-driven model-free adaptive sliding mode control (MFASMC) approach to address the Haidou-1 ARV motion control problem with uncertainties. However, some concerns are still on the way.

1. The authors claim to propose a data-driven control approach. However, the detailed design is missing. Please provide additional details.

2. The contribution of this paper is unclear. As far as I am aware, learning-based AUV control is not a novel concept, as discussed in [1].

3. It would be beneficial to enhance readability by adding some general applications of the AUV in the background section, such as data collection assisted by AUVs [2][3]. Because not everyone knows the AUVs value.

4. The figures in the simulation section are low-resolution. Please replace them with high-resolution versions.

[1] M. J. Er, H. Gong, Y. Liu and T. Liu, "Intelligent Trajectory Tracking and Formation Control of Underactuated Autonomous Underwater Vehicles: A Critical Review," in IEEE Transactions on Systems, Man, and Cybernetics: Systems, vol. 54, no. 1, pp. 543-555, Jan. 2024, doi: 10.1109/TSMC.2023.3312268.

[2] X. Hou, J. Wang, T. Bai, Y. Deng, Y. Ren and L. Hanzo, "Environment-Aware AUV Trajectory Design and Resource Management for Multi-Tier Underwater Computing," in IEEE Journal on Selected Areas in Communications, vol. 41, no. 2, pp. 474-490, Feb. 2023, doi: 10.1109/JSAC.2022.3227103.

[3] M. Huang, K. Zhang, Z. Zeng, T. Wang and Y. Liu, "An AUV-Assisted Data Gathering Scheme Based on Clustering and Matrix Completion for Smart Ocean," in IEEE Internet of Things Journal, vol. 7, no. 10, pp. 9904-9918, Oct. 2020, doi: 10.1109/JIOT.2020.2988035.

Comments on the Quality of English Language

Need to be improved

Round 2

Reviewer 1 Report

Comments and Suggestions for Authors

I have read the paper and I am okay with that. Some equations are still too wordy, please consider making them concise.

Reviewer 2 Report

Comments and Suggestions for Authors

The authors have addressed all my concerns. I have no further comments. 
